# The Alpha-1 Subunit of the Na^+^/K^+^-ATPase (ATP1A1) Is a Host Factor Involved in the Attachment of Porcine Epidemic Diarrhea Virus

**DOI:** 10.3390/ijms24044000

**Published:** 2023-02-16

**Authors:** Moukang Xiong, Xianhui Liu, Tairun Liang, Yanfang Ban, Yanling Liu, Leyi Zhang, Zheng Xu, Changxu Song

**Affiliations:** 1College of Animal Science & National Engineering Center for Swine Breeding Industry, South China Agriculture University, Guangzhou 510642, China; 2Lingnan Modern Agricultural Science and Technology Guangdong Laboratory, Guangzhou 510000, China

**Keywords:** PEDV, ATP1A1, attachment, S1 protein, replication

## Abstract

Porcine epidemic diarrhea (PED) is an acute and severe atrophic enteritis caused by porcine epidemic diarrhea virus (PEDV) that infects pigs and makes huge economic losses to the global swine industry. Previously, researchers have believed that porcine aminopeptidase-N (pAPN) was the primary receptor for PEDV, but it has been found that PEDV can infect pAPN knockout pigs. Currently, the functional receptor for PEDV remains unspecified. In the present study, we performed virus overlay protein binding assay (VOPBA), found that ATP1A1 was the highest scoring protein in the mass spectrometry results, and confirmed that the CT structural domain of ATP1A1 interacts with PEDV S1. First, we investigated the effect of ATP1A1 on PEDV replication. Inhibition of hosts ATP1A1 protein expression using small interfering RNA (siRNAs) significantly reduced the cells susceptibility to PEDV. The ATP1A1-specific inhibitors Ouabain (a cardiac steroid) and PST2238 (a digitalis toxin derivative), which specifically bind ATP1A1, could block the ATP1A1 protein internalization and degradation, and consequently reduce the infection rate of host cells by PEDV significantly. Additionally, as expected, overexpression of ATP1A1 notably enhanced PEDV infection. Next, we observed that PEDV infection of target cells resulted in upregulation of ATP1A1 at the mRNA and protein levels. Furthermore, we found that the host protein ATP1A1 was involved in PEDV attachment and co-localized with PEDV S1 protein in the early stage of infection. In addition, pretreatment of IPEC-J2 and Vero-E6 cells with ATP1A1 mAb significantly reduced PEDV attachment. Our observations provided a perspective on identifying key factors in PEDV infection, and may provide valuable targets for PEDV infection, PEDV functional receptor, related pathogenesis, and the development of new antiviral drugs.

## 1. Introduction

Porcine epidemic diarrhea virus is a single-stranded positive sense envelope membrane RNA virus belonging to the genus alphacoronavirus of the coronaviridae family [1]; it is transmitted mainly by fecal-oral transmission [2]. PEDV has two untranslated regions (UTR) at the 5′N-terminal and 3′C-terminal end and seven open reading frames (ORF1a, ORF1b, S, ORF3, E, M and N). In addition to the auxiliary protein ORF3, other proteins including S, E, M and N constitute the viral structural proteins [3]. S protein is a glycoprotein on the surface of the virus, and has the size of approximately 200 kDa with a neutralizing antibody epitope and a receptor binding site involved in recognition [4]. Two different conformations exist in the S protein: the pre-fusion conformation is a clove-shaped trimer consisting of three individual S1 heads and a trimeric S2 stalk; the post-fusion conformation is a trimeric S2 [5,6]. Understanding the basic mechanisms of PEDV-host interactions and clarifying the role of host factors in PEDV infection can play a vital part in the development of new antiviral drugs and effective broad-spectrum vaccines.

Recognition of the virus by the cellular receptor is a crucial step in the virus life cycle; it enables the virus to enter and infect host cells. According to existing reports, transgenic mice with pAPN could be infected with PEDV [7]. PEDV S1 has been biochemically interacted with soluble pAPN in a dot-blot assay [8]. Hence, pAPN has been commonly described as a functional receptor for PEDV entry into cells [9]. However, it was shown that IPEC-J2 cells with low expression of pAPN did not affect PEDV infection and that PEDV was able to infect Vero-E6 cells that did not express pAPN [10]. pAPN knockout pigs still remained susceptible to PEDV [11]. These results suggest that pAPN′s role as a functional receptor for PEDV needs further study.

Virus overlay protein binding assay (VOPBA) is a common method of screening viral receptors, using the method and principle of protein immunoblotting to screen for host proteins that bind to viruses by combining the specific interaction of viral protein and host protein [12]. Many virus-binding proteins have been reported to be identified using VOPBA. Norwalk virus (NV) interacted with the NV attachment (NORVA) protein to trigger the viral attachment [13]. Respiratory Syncytial Virus (RSV) was involved in the binding of nucleolin [14]. Grouper heat-shock cognate protein 70 (GHSC70) interacted with nervous necrosis virus (NNV) capsid protein to benefit the NNV attachment [15].

Na+/K+-ATPase (NKA) is a channel protein embedded in the phospholipid bilayer of the cell membrane, which has physiological functions such as ATPase activity and maintenance of intracellular and extracellular osmotic pressure [16,17,18]. NKA consists of four α subunits, three β subunits and γ isoforms. The N and C termini of the α-subunit are intracellular and anchored to the plasma membrane through 10 transmembrane helical regions to form ion channels [19]. The α1 isoform (ATP1A1) is widely expressed in eukaryotic cells. In addition, ATP1A1 has been found to be overexpressed in a variety of cancer cells, such as breast cancer, liver cancer and glioma [20,21,22,23,24]. ATP1A1 has not only played an important role in cancer cells, but also participated in different stages of viral infection, including viral attachment [25,26] and replication [27].

In this study, VOPBA was performed for the screening of ATP1A1 binding protein. Subsequently, we investigated the relationship between the host protein ATP1A1 and PEDV infection for the first time using IPEC-J2 and Vero-E6 cells. We found that PEDV infection of host cells upregulated the expression of ATP1A1 to facilitate PEDV infection. Further analysis showed that ATP1A1 is a host factor that facilitates PEDV attachment to the host cells.

## 2. Results

### 2.1. ATP1A1 CT Structural Domain Is Required for Interaction with PEDV S1

VOPBA is a common method of screening viral receptors [12,28,29]. We performed a VOPBA, immunoprecipitation with S1 monoclonal antibody (mAb) to PEDV, and after mass spectrometry analysis (data not shown), we screened ATP1A1 binding proteins with wide distribution, high abundance and presence of cell membranes for subsequent experiments.

To confirm that the PEDV S1 protein interacts with the host protein ATP1A1, IP analysis was performed on IPEC-J2 cells using PEDV S1 and ATP1A1 mAb. The results showed that S1 protein interacted with the endogenous ATP1A1 (Figure 1A). To identify which structural domain of ATP1A1 is responsible for the interaction with PEDV S1 protein, and to predict the structural domain using the ATP1A1 amino acid sequence at the HMMER website, we constructed the truncated plasmids of all structural domains of ATP1A1 as shown in Figure 1B. They transfected separately into Vero-E6 and then infected PEDV for IP analysis. We found that only the full-length ATP1A1 could interact with S1 (Figure 1C). To clarify whether it is the CT structural domain or only the full-length ATP1A1 that can interact with S1, we constructed plasmids expressing only the CT structural domain for analysis. Further analysis showed that the ATP1A1 CT structural domain is required for its binding to PEDV S1 (Figure 1D). These findings demonstrated that the ATP1A1 CT structural domain is required for interaction with the PEDV S1 protein. 

### 2.2. Knockdown of ATP1A1 Expression by siRNAs Transfection

To understand whether ATP1A1 is involved in PEDV infection, we synthesized two specific small interfering RNAs (siRNAs) of porcine and monkey origin, respectively, to test their functions. When cells were transfected separately with ATP1A1-specific siRNAs (siRNA-ATP1A1-A/B or siRNA-mATP1A1-A/B) for 48 h (no toxicity to cells), it was found that siRNA-ATP1A1-A/B or siRNA-mATP1A1-A/B significantly downregulated ATP1A1 mRNA transcription and protein expression in both IPEC-J2 and Vero-E6 cells; while, in the control groups (siRNA-NC or siRNA-mNC), high ATP1A1 expression could be detected. Considering its overall efficiency in the target cells, the siRNA-ATP1A1-A and siRNA-mATP1A1-A were therefore selected for subsequent experiments (Figure 2A–F).

### 2.3. Knockdown of Endogenous ATP1A1 Expression Suppresses PEDV Infection

First, we investigated the biological significance of knocking down the expression of ATP1A1 in PEDV-infected target cells. Twenty-four hours after siRNAs transfection, IPEC-J2 and Vero-E6 cells were infected with PEDV, and samples were collected for PEDV viral load determination. PEDV N mRNA levels significantly reduced after PEDV infection (Figure 3A,B). Western blot results indicated that the PEDV N protein decreased in IPEC-J2 and Vero-E6 cells compared with that of the NC groups (Figure 3C,D). In addition, TCID_50_ assays performed in IPEC-J2 and Vero-E6 cells after reduced expression of endogenous ATP1A1 showed decreased PEDV titers (Figure 3E,F). As shown in Figure 3G,H, a reduction in the amount of PEDV fluorescence was observed in IPEC-J2 and Vero-E6 cells transfected with ATP1A1-specific siRNAs compared with that of the NC groups. Collectively, these data suggested that knockdown of endogenous ATP1A1 results in reduced PEDV replication in target cells. 

### 2.4. NKA Inhibitors Promote Degradation of ATP1A1 and Effectively Reduce PEDV Infection

Ouabain is one of the Cardiotonic steroid (CTS) drugs; it was reported that the binding of Ouabain to NKA leads to a change in its protein conformation and the internalization of ATP1A1 into the cytoplasm to participate in lysosomal-mediated degradation [30]. PST2238 (digitalis toxin derivative) is used as a competitive inhibitor of Ouabain. Therefore, we tested the effect of Ouabain and PST2238 on PEDV infection by targeting NKA.

The cytotoxicity of Ouabain and PST2238 drugs were measured in IPEC-J2 cells after serial dilutions, and we chose concentrations of Ouabain (1 nM) and PST2238 (1 uM) which had no effect on cell viability after 48 h of treatment (Figure 4A,B). First, we infected IPEC-J2 cells with PEDV after pretreatment with Ouabain or PST2238 at non-cytotoxic concentrations for 1 h, and collected infected cells and culture supernatants for PEDV viral load assay. We found that drug pretreatment notably inhibited PEDV replication (Figure 4C). We pretreated IPEC-J2 cells with a gradient dilution of Ouabain or PST2238 for 1 h before infection with PEDV. We found that Ouabain and PST2238 significantly reduced PEDV RNA in a dose-dependent manner (Figure 4D). We then examined the changes in ATP1A1 and PEDV-N protein levels after drugs treatment. The ATP1A1 and PEDV N protein expression decreased gradually in a dose-dependent manner as the drugs concentrations of Ouabain or PST2238 were increased (Figure 4E,F). Inhibitors of Ouabain and PST2238 inhibited PEDV replication in a dose-dependent manner, according to the IFA data (Figure 4G). These results suggested that NKA inhibitors are effective in reducing PEDV infection.

### 2.5. Overexpression of ATP1A1 Promotes PEDV Infection

To clarify what role the ATP1A1 protein plays in PEDV infection, we overexpressed the ATP1A1 in PEDV-infected IPEC-J2 and Vero-E6 cells, and then detected viral susceptibility. First, we found that overexpression of ATP1A1 significantly promoted PEDV N mRNA level compared with that of the empty vector transfection group (Figure 5A). We also examined the effect of overexpression of ATP1A1 on PEDV N protein levels by Western blot. In addition to using the PEDV G1 genotype strain CV777, we also performed tests using the G2 strain GDgh isolated in our laboratory. Consistent with the mRNA results, overexpression of ATP1A1 significantly increased the expression of PEDV N protein (Figure 5C). Meanwhile, ATP1A1 protein significantly promoted the expression of PEDV N protein in a dose-dependent manner (Figure 5E). The viral titer in the supernatant of IPEC-J2 cells overexpressing ATP1A1 was higher than that of cells transfected with empty vector (Figure 5G). In the immunofluorescence assay, the transfection of exogenous ATP1A1 into IPEC-J2 cells resulted in increased amount of fluorescence of PEDV compared with the empty vector group (Figure 5I). We also performed the experiments where PEDV infected Vero-E6 cells overexpressing ATP1A1, and viral yield was measured by qPCR, Western blot, IFA and TCID_50_. The results showed that overexpression of ATP1A1 in Vero-E6 cells facilitated the replication of PEDV (Figure 5B,D,H,J). In addition, we attempted to overexpress ATP1A1 protein on PTR2 and DF-1 cells which could not support the infection of PEDV (Figure 5F). These results indicated that overexpression of ATP1A1 protein makes target cells more susceptible to PEDV infection, although it does not convert uninfected cell lines to be susceptible, which also suggested that ATP1A1 protein plays an important role in viral infection.

### 2.6. PEDV Infection Upregulates ATP1A1 Protein Expression in Target Cells

To understand the association between the host protein ATP1A1 and PEDV infection, we analyzed the changes in mRNA and protein levels of ATP1A1 after PEDV infection. As determined by qPCR, PEDV infection resulted in a significant upregulation of ATP1A1 mRNA levels (Figure 6A,B). Similarly, the protein levels of ATP1A1 increased after PEDV infection, consistent with that of the mRNA results (Figure 6C,D).

To further understand the association between ATP1A1 and PEDV infection, we used immunofluorescence to observe the expression of ATP1A1 protein in target cells after PEDV infection. The results of immunofluorescence experiments revealed that enhanced fluorescence of ATP1A1 proteins occurred after PEDV infection (Figure 6E). To ensure that the phenomenon was not restricted to IPEC-J2 cells, we repeated the experiment in Vero-E6 cells as well. Consistent with the phenomenon on IPEC-J2, ATP1A1 protein was also enhanced in PEDV-infected Vero-E6 cells compared with mock-treated Vero-E6 cells (Figure 6F). These data indicated that PEDV infection induces increased expression of ATP1A1 protein in target cells, suggesting that the expression of ATP1A1 protein may be related to PEDV infection.

### 2.7. Knockdown of ATP1A1 Affects the Attachment of PEDV

According to the proportion of homologous amino acids with other coronavirus S proteins, PEDV S protein can be divided into S1 and S2 structural domains, and the S1 protein plays a crucial role in recognition of viral particles with host proteins [31]. We screened the ATP1A1 protein from the mass spectrometry results which interacted with the PEDV S1 protein (Figure 1A), and the viral S1 protein is mainly involved in viral attachment.

We first explored the effect of downregulation of ATP1A1 protein expression on PEDV attachment, which we did for G1 genotype CV777 and G2 genotype GDgh of PEDV. We found that downregulation of ATP1A1 expression significantly inhibited the attachment and internalization processes of G1 type and G2 type PEDV (Figure 7A). We speculated that the host protein ATP1A1 may be involved in the attachment of PEDV to target cells, and it may be a host factor that promotes PEDV attachment. The downregulated ATP1A1 expression also has an effect on the internalization phase, and we speculated that it may also be involved in the internalization of PEDV, and we will investigate whether ATP1A1 is involved in internalization in the future. 

In addition, we used flow cytometry to quantify the interaction between ATP1A1 and cell surface PEDV particles. The number of positive cells bound by ATP1A1 to the viral particle S1 protein gradually increased at 60 min of infection compared to that of 0 min of infection; at 120 min of infection, the number of positive cells bound to the cell surface gradually decreased as the viral particles entered the cells and was less than the number of positive cells at the time of initial infection (Figure 7B,C). We speculated that ATP1A1 may be involved in the attachment stage of PEDV infection.

### 2.8. The Host Protein ATP1A1 Co-Localizes with the PEDV S1 Protein Early in PEDV Infection

To simulate the biological process of virus infection of cells, cells were infected with PEDV and immunofluorescence staining with ATP1A1 and PEDV S1 mAbs. We observed the co-localization of PEDV S1 and ATP1A1 protein in the early stages of infection. ATP1A1 co-localized with PEDV in target cells (Figure 8A,B). At the time of infection of 1 h, the ATP1A1 and S1 co-localization phenomenon is more obvious on Vero-E6 than IPEC-J2 cells (Figure 8C,D).

### 2.9. Monoclonal Antibody Pretreatment of ATP1A1 Effectively Inhibits PEDV Attachment

To confirm the role of ATP1A1 in PEDV attachment, monoclonal antibody (mAb) against ATP1A1 was incubated with IPEC-J2 and Vero-E6 cells to interfere with the interaction between ATP1A1 and PEDV. ATP1A1 mAb pretreatment decreased PEDV RNA abundance in target cells in a dose-dependent manner compared with the DMEM group (Figure 9A,B). The IPEC-J2 and Vero-E6 cells were preincubated with ATP1A1 mAb by serial dilution, and then infected with PEDV; in the end, the cells were collected for Western blotting analysis. PEDV N protein expression was significantly reduced in the group of ATP1A1 mAb pretreatment, and importantly with a dose dependent manner (Figure 9C,D). Pre-incubation of IPEC-J2 and Vero-E6 cells with a 16,000-fold dilution of ATP1A1 mAb was followed by infection with PEDV, and a significant reduction in the daughter virus was demonstrated by TCID_50_ assay (Figure 9E,F) and IFA data (Figure 9G,H). These results suggested that ATP1A1 mAb could be able to significantly block the PEDV attachment.

## 3. Discussion

PEDV caused acute infectious and severe atrophic enteritis of the small intestinal villi in infected pigs, leading to severe vomiting and diarrhea, dehydration, loss of appetite and depression [32]. Due to the devastating impact on the global pig industry and the potential threat posed across species by PEDV, understanding the interaction between this virus and its host is urgent for the infection mechanism and the development of antiviral strategies.

The binding of the virus to receptors on the cell surface is the first step in virus-infected cells [33], and PEDV invades host cells through membrane fusion. Previous studies reported that PEDV utilizes heparan sulfate on the cell surface to facilitate the attachment of host cells [34], sialic acid is beneficial to PEDV binding and entry [35,36], the presence of cholesterol in cell membranes is required for PEDV entry into cells [37], transferrin receptor 1 on the cell surface can increase the susceptibility of PEDV to piglets [38] and the tight junction protein Occludin facilitates PEDV entry [39]. Furthermore, integrin αvβ3 is involved in the cellular uptake of porcine intestinal α-coronavirus [40], and also enhances PEDV replication in Vero-E6 and IPEC-J2 cells [41]. Previously, porcine aminopeptidase-N (pAPN) was widely accepted as a functional receptor for PEDV [9,42]. Further studies have shown that changes in pAPN enzyme activity are factors affecting PEDV infection [10]. In addition, surface plasmon resonance results indicated that the pAPN extracellular structural domain did not interact with PEDV S1 or S2 proteins [43]. These different experimental results suggested that pAPN is questionable as a true functional receptor for PEDV and that there may be other receptors that facilitate the PEDV attachment of host cells.

Na^+^/K^+^-ATPase (NKA) is an energy exchange ion pump first proposed by Skou in 1957 [44]. NKA plays an important role in active transport, energy metabolism and signaling [36,45,46,47]. Recent studies have shown that NKA is participating in cell signaling pathways that are not dependent on ion pump function [48]. The signal transduction role is mainly attributed to the α subunit [49]. Ouabain as a specific inhibitor of NKA can trigger signal transduction without affecting the sodium–potassium pump function or ion homeostasis [50], which has been reported to inhibit viral replication including herpes simplex virus 1 [51,52] and adenovirus [53]. PST2238 inhibits Ouabain binding and signal transduction [54]. PST2238 also inhibits viral replication and prevents the entry of human respiratory syncytial virus (RSV) by inhibiting ATP1A1 activation [26]. In addition, ATP1A1 has been associated with various viral infections, for instance, SARS-CoV-2 virus RNA has interacted with host protein ATP1A1 during infection [55]. ATP1A1 has played an important role as a host factor in SARS-CoV-2 infection, and inhibition of ATP1A1 expression blocked fetal intestinal infection [56]. Moreover, the Ebola VP24 protein has interacted with ATP1A1 and treatment with ATP1A1 inhibitor Ouabain reduced viral infection [57].

In our study, we found that the CT domain of ATP1A1 interacts with PEDV S1 protein (Figure 1). In addition, we showed that downregulating the expression of ATP1A1 with siRNAs reduced the PEDV infection, but the viral suppression effect in the Vero-E6 cell line was not as pronounced as that of IPEC-J2, which we speculate may be due to the two cell lines being so different, even with different genera belongings (Figure 3).When we pretreated the target cells with drugs, a significant reduction in PEDV replication could be observed (Figure 4). Moreover, we observed that the overexpression of ATP1A1 could promote PEDV infection (Figure 5). We also demonstrated that PEDV infection induced massive ATP1A1 expression, which may facilitate the PEDV infection in host cells (Figure 6). These results suggested that the ATP1A1 protein plays an important role in PEDV infection. The surface spike (S) protein on coronaviruses is a type I glycoprotein, consisting of S1 receptor-binding domain and S2 membrane fusion domain, which is a determinant of the virus tropism [58,59]. The S protein has a crucial role in binding to host receptors during the attachment phase and in mediating membrane fusion during the invasion phase [8,60]. Therefore, we first investigated what role ATP1A1 plays in the PEDV attachment phase. The results showed that ATP1A1 was involved in the attachment phase of PEDV (Figure 7) and co-localized with S1 protein (Figure 8). Pretreatment with anti-ATP1A1 mAb significantly inhibited PEDV attachment, showing that ATP1A1 contributed to PEDV attachment (Figure 9). Collectively, these results detailed the mechanism that ATP1A1 affects PEDV replication, which may act during the attachment phase of viral infection to host cells, providing a basis of the development of novel antiviral drugs.

Based on the above experimental results, we present a pattern diagram to better describe the role of ATP1A1 in PEDV attachment (Figure 10). Besides the currently unknown functional receptor, the PEDV S1 protein also binds to ATP1A1 at the cell membrane surface which facilitates the PEDV attachment to the host cells.

In conclusion, our findings suggested that ATP1A1 may be a host factor that facilitates PEDV attachment, which may be also applied to the understanding of the interaction between other coronaviruses and hosts as well. Additionally, ATP1A1 is widely distributed and abundantly present on the cell membrane surface, which explains the extensive cytohagocytosis of PEDV well.

## 4. Materials and Methods

### 4.1. Cells and Viruses

IPEC-J2, Vero-E6, and Vero cells were separately cultured at 37 °C in a humidified incubator with a 5% CO_2_ atmosphere in Dulbecco’s minimum essential medium (DMEM; Procell, Wuhan, China) supplemented with 10% fetal bovine serum (FBS; Procell, China) and penicillin (100 U/mL; NCM, Suzhou, China) and streptomycin (100 ug/mL; NCM, China). The PEDV strain CV777 with genotype 1 (GenBank accession no. LT906620) and PEDV strain GDgh with genotype 2 (GenBank accession no. MG983755) were isolated and preserved at the South China Agricultural University, Guangzhou, China [61]. In addition to the experimentally indicated strain subtypes, the PEDV strain used was genotype 2 strain GDgh. The viral infection dose is indicated in the figure legends.

### 4.2. Antibodies, Inhibitors, and Reagents

Rabbit anti-ATP1A1 mAb (catalog no. GY5154) was purchased from AbWays (Shanghai, China). CoraLite488-conjugated Goat Anti-Mouse IgG (catalog no. SA00013-1), CoraLite488-conjugated Goat Anti-Rabbit IgG (catalog no. SA00013-2), CoraLite594-conjugated Goat Anti-Mouse IgG (catalog no. SA00013-3), CoraLite594-conjugated Goat Anti-Rabbit IgG (catalog no. SA00013-4) and Mouse anti-GAPDH mAb (catalog no. 60004-1-Ig) were purchased from Proteintech (Wuhan, China). Mouse anti-PEDV-N mAb (catalog no. M100048) was purchased from Zoonogen (Beijing, China). Mouse anti-FLAG mAb (catalog no. F1804-200UG) was purchased from Sigma (St. Louis, MI, USA). Goat Anti-Rabbit IgG (catalog no. ab6721) and Goat Anti-Mouse IgG (catalog no. ab205719) were purchased from Abcam (Cambridge, UK). The PEDV S1 mAb was a generous gift from Professor Kegong Tian of Henan Agricultural University, China. In Western blot experiments, the primary antibody was diluted 1:1000 and the secondary antibody was diluted 1:5000. In the IFA assay, the antibody was diluted 1:300.

Ouabain (catalog no. HY-B0542) and PST2238 (catalog no. HY-12283) were purchased from MCE (Shanghai, China).

Lipofectamine RNAiMAX transfection reagent (catalog no. 13778150) was purchased from Invitrogen. FuGENE transfection reagent was purchased from Promega (Madison, WI, USA). Cell lysis buffer for Western and IP (catalog no. P0013, Beyotime, China) was purchased from Beyotime (Shanghai, China). Cell Counting Kit-8 was purchased from YEASEN (Shanghai, China).

### 4.3. Plasmid Constructs

The four truncated as well as full-length ATP1A1 genes were amplified from small intestinal epithelial cells using the primers in Table 1 and cloned into the eukaryotic expression vector pECMV-3×FLAG-N for mammalian cell expression. Nucleotide sequences of the construct plasmids were compared to ensure that the correct clones were used in this study.

### 4.4. Western Blot and IP

Cells were washed with PBS and incubated in WB lysate containing protease inhibitor (catalog no. GK10014; GLPBIO, Montclair, CA, USA) on ice to inhibit protein degradation. Samples were separated and transferred to polyvinylidene fluoride (PVDF) membranes (catalog no. ISEQ00010; Merck Millipore, Darmstadt, Germany). Skim milk 5% with PVDF membrane were blocked at room temperature for 1 h, followed by overnight incubation with primary antibody dilutions at 4 °C. The PVDF membrane was washed five times with PBST (PBS with 0.05% Tween 20) and incubated with secondary antibody at room temperature for 1 h, and protein bands were detected using BeyoECL Plus (catalog no. P0018M, Beyotime, China).

FLAG, FLAG-A1, FLAG-A2, FLAG-A3, FLAG-A4 or FLAG-ATP1A1 expressed plasmid were transfected, respectively, into Vero-E6 cells for 24 h, and the cells were collected 24 h after PEDV at 0.1 MOI infection. Samples were incubated with anti-S1-mAb overnight at 4 °C and then incubated with Pierce Protein A/G Magnetic Beads for 1 h at 4 °C [62]. Samples were washed 3 times with PBS, and protein samples were prepared and detected by Western blot using the indicated antibodies.

### 4.5. RNA Interference

The siRNAs against porcine-derived and monkey-derived ATP1A1, siRNA-negative control (siRNA-NC or siRNA-mNC) were designed and synthesized by Sangon Biotech (Shanghai, China). The indicated siRNAs were introduced into the cells with RNAiMAX (Invitrogen) reagent at a concentration of 50 nM according to the manufacturer’s instructions. Forty-eight hours after transfection, the cells were scraped and assayed by quantitative RT-PCR and Western blotting for specific gene silencing. In some experiments, cells were transfected for 24 h and infected with PEDV at an MOI of 0.1 for subsequent experiments. The indicated siRNAs are listed in Table 2.

### 4.6. Cell Viability Detection

Cell viability was detected by a cell counting kit-8 (CCK-8). Cells were treated with the specified concentration of inhibitors for 1 h at 37 °C or transfected with siRNAs for 24 h at 37 °C. After adding CCK-8 solution and incubating at 37 °C for 2 h, the absorbance at 450 nm was measured using an enzyme marker (Gene, South San Francisco, CA, USA).

### 4.7. Inhibitor Treatments

IPEC-J2 cells were co-incubated with non-cytotoxicity specific inhibitor or DMSO mixed with PEDV for 1 h. They were changed into 2% maintenance solution containing the same concentration of inhibitor or dimethyl sulfoxide (DMSO) and incubated for the indicated time for subsequent experiments [26].

### 4.8. ATP1A1 mAb Inhibition Assay

Based on previous studies, we examined the effect of ATP1A1 mAb on PEDV infection [63]. IPEC-J2 and Vero-E6 cells were incubated with ATP1A1 mAb at required dilution in DMEM at 37 °C for 1 h. They were incubated with DMEM containing the corresponding antibodies and PEDV (0.1 MOI) strain GDgh with genotype 2 at 4 °C for 1 h. After washing the cells three times with PBS, the cells were incubated again with antibodies contained in the appropriate concentrations at 37 °C, and the cells were collected and assayed at the indicated times.

### 4.9. Quantitative Real-Time PCR (RT-qPCR)

Total RNA was extracted using HiPure Total RNA Mini Kit (catalog no. R4111-03; Magen, Guangzhou, China), and cDNA was produced by reverse transcription using Evo M-MLV RT Premix (catalog no. AG11706; AG, Changsha, China). The cDNAs from different samples were amplified by RT-qPCR to measure the target gene. The RT-qPCR was performed using Eastep qPCR Master Mix (catalog no. LS2062; Promega, Madison, WI, USA) on QuantStudio 5 (Thermo Fisher Scientific, Waltham, MA, USA) and programmed as follows: 95 °C for 2 min (1 cycle), 95 °C for 15 s and 60 °C for 60 s (40 cycles). The primers were listed in Table 2. Relative quantification was determined by the 2 (-Delta Delta CT) Method [64].

### 4.10. Immunofluorescence Assay and Confocal Microscopy

Cells are grown on culture plates or slides and processed as desired for the experiment. Then, 4% paraformaldehyde was fixed at 4 °C for 1 h, 0.5% Triton X-100 penetrated at 37 °C for 10 min, and 1% BSA blocking buffer was used for closure to reduce non-specific binding. Samples were incubated with primary and secondary antibodies as specified. The staining of cell nuclei was performed using DAPI staining solution (DAPI, catalog no. c1006; Beyotime, China) and observed via fluorescence microscopy (DS-Qi2, Nikon, Japan) with the confocal microscopy (AX, Nikon, Japan).

### 4.11. Flow Cytometry

To measure the number of positive cells of ATP1A1 binding to viral particles on the cell surface, cells were cultured under normal conditions to a density of 90%. IPEC-J2 or Vero-E6 cells were chilled on ice for 10 min to synchronize PEDV infection, and the growth medium was replaced with PEDV virus medium containing an MOI of 1. The cells were incubated on ice for 1 h to synchronize infection and then transferred to a 37 °C incubator for the indicated time. The cells were washed 3 times with PBS, digested with trypsin and collected, washed twice with pre-cooled PBS, centrifuged and precipitated, the supernatant was discarded and the cells were resuspended with PBS and counted. Then, 2 μL of PEDV-S1 and ATP1A1 antibodies were added to each tube and incubated for 60 min at 4 °C avoiding light, then washed twice with PBS. Next, we added secondary antibody and incubated for 30 min at 4 °C, avoiding light. The cells were washed twice with PBS and resuspended in PBS for flow cytometric detection (catalog no. FACS101; BD, Franklin Lakes, NJ, USA).

### 4.12. TCID_50_ Assay

PEDV was inoculated at an MOI of 0.1 into cells treated according to experimental requirements, incubated for 1 h and then washed with PBS. At 24 h, the titer of the offspring virus was determined according to the method of Reed and Muench [65]. Briefly, 0.1 mL of 10-fold gradient diluted (10^−7^ to 10^−1^) sample was added to the Vero-E6 cells in the 96-well plate. After six days, the cytopathic effects (CPEs) were observed with an inverted microscope (ECLIPSE TS100, Japan) and the number of CPEs wells was counted.

### 4.13. Statistical Analysis

All data were expressed as the means ± standard deviations (SD) and analyzed by Student’s *t* test using GraphPad Prism software (version 8.0). Values of *p* < 0.05 were considered to be statistically significant and were indicated as follows: *, *p* < 0.05; **, *p* < 0.01; ***, *p* < 0.001.

## Figures and Tables

**Figure 1 ijms-24-04000-f001:**
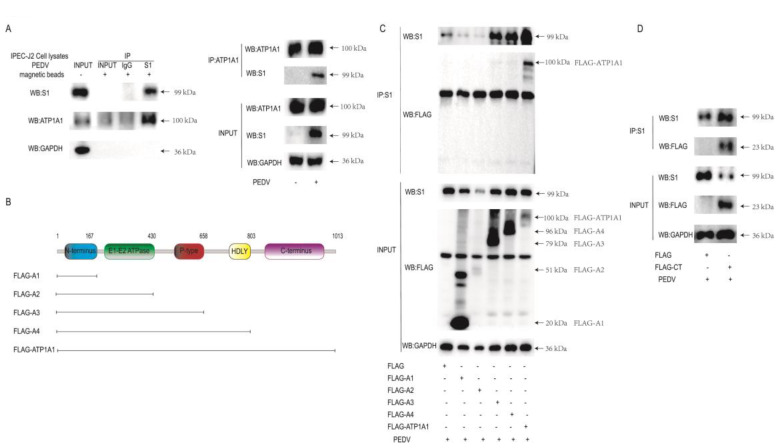
The ATP1A1 CT structural domain is required for the interaction with PEDV S1. (**A**) PEDV S1 interacts with endogenous ATP1A1 protein. PEDV at 0.1 MOI infected IPEC-J2 cells for 36 h. Western blot of CO-IP with mouse anti-S1 mAb and rabbit anti-ATP1A1 mAb. (**B**) The schematic diagram of the truncated structural domains of ATP1A1. (**C**) Full-length ATP1A1 interacted with S1 protein. Full-length ATP1A1, truncated A1, A2, A3 and A4 structural domains were transfected, fy, to Vero-E6 cells for 24 h and then PEDV at 0.1 MOI infected cells for 24 h. Western blot of co-immunoprecipitations from lysates with mouse anti-S1 mAb and mouse anti-FLAG mAb. (**D**) ATP1A1 CT domain interacted with S1 protein. Vero-E6 cells were transfected with the plasmid expressing ATP1A1-CT for 24 h and were infected with PEDV at 0.1 MOI for 24 h. Western blot of co-immunoprecipitations from lysates with mouse anti-S1 mAb and mouse anti-FLAG mAb.

**Figure 2 ijms-24-04000-f002:**
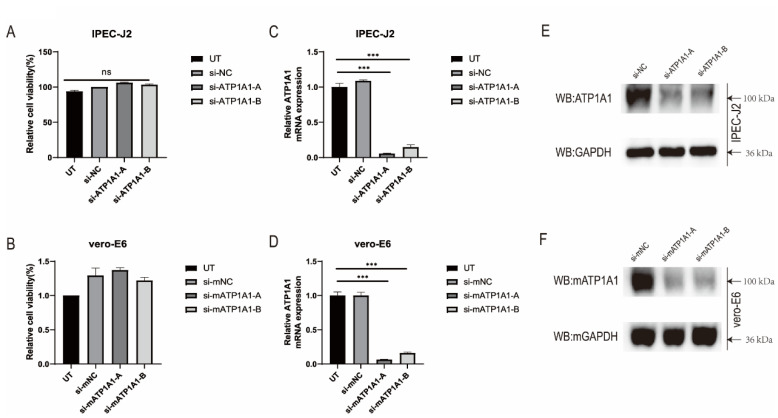
ATP1A1 knockdown by siRNAs transfection. (**A**,**B**) The cytotoxicity of siRNAs. IPEC-J2 and Vero-E6 cells were transfected with siRNAs against ATP1A1 for 48 h and analyzed using the CCK-8 kit. Data represent means ± SD from three independent experiments. ns, no significant difference. (**C**,**D**) Relative quantification of ATP1A1 mRNA. Cells were transfected with siRNAs against ATP1A1 for 48 h. Total cellular RNA was extracted, reverse transcribed, and quantified. The siRNA-NC or siRNA-mNC assigned the value of 1.0, data represent means ± SD from three independent experiments. ***, *p* < 0.001. (**E**,**F**) Western blot analysis of ATP1A1 protein expression. The cells were lysed after 48 h of interference and Western blot was performed using rabbit anti-ATP1A1 mAb and mouse anti-GAPDH mAb. Detection using the corresponding secondary antibody.

**Figure 3 ijms-24-04000-f003:**
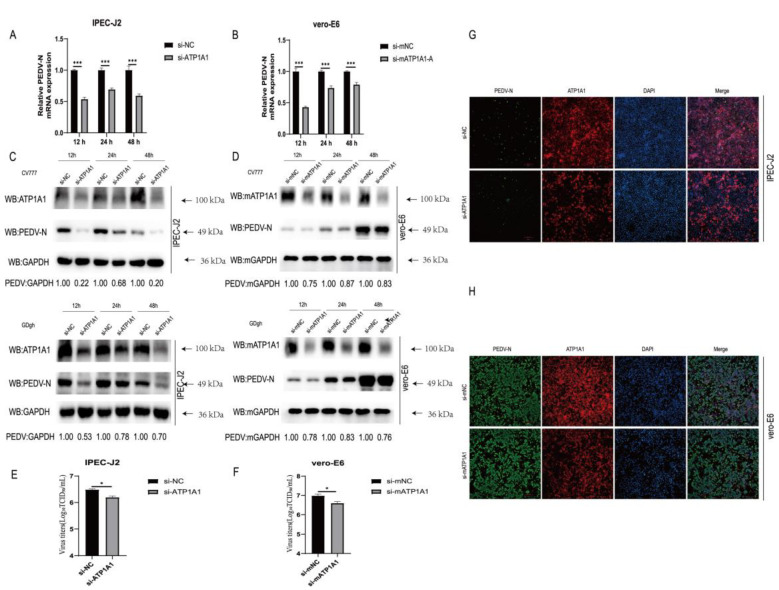
Knockdown of endogenous ATP1A1 expression suppresses PEDV infection. (**A**,**B**) Knockdown of endogenous ATP1A1 significantly suppresses PEDV RNA abundance. Transfected with siRNAs against ATP1A1 or siRNA-NC for 24 h, cells were collected at 12, 24 and 48 h after PEDV (0.1 MOI) infection of target cells for quantitative PCR assay. Data represent means ± SD from three independent experiments. ***, *p* < 0.001. (**C**,**D**) Knockdown of endogenous ATP1A1 reduces in PEDV N protein expression. After knocking down with the expression of ATP1A1 for 24 h, target cells were infected with PEDV (CV777-G1 or GDgh-G2) at 0.1 MOI and collected and lysed at different time points for Western blot analysis. (**E**,**F**) Knockdown of endogenous ATP1A1 suppresses PEDV viral titers. PEDV at 0.1 MOI infected cells with low ATP1A1 expression for 24 h, and cell supernatants were extracted for TCID_50_ assay. Data represent means ± SD from three independent experiments. *, *p* < 0.05. (**G**,**H**) Knockdown of endogenous ATP1A1 reduces PEDV infection. Cells were transfected with siRNAs for 24 h, and then PEDV (0.1 MOI) infected for 24 h. The cells were stained with anti-ATP1A1 mAb (red) and anti-PEDV-N mAb (green). Staining of cell nuclei using DAPI (blue) staining solution. The processed samples were photographed and analyzed using fluorescence microscopy. Scale bars, 200 μm.

**Figure 4 ijms-24-04000-f004:**
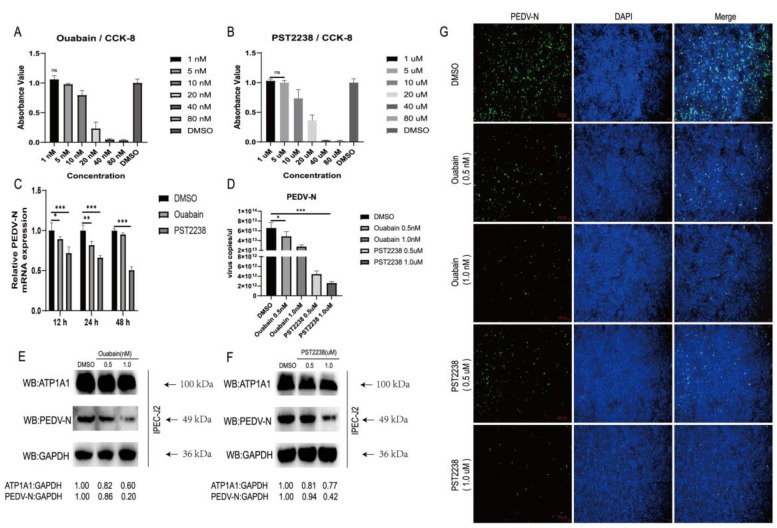
NKA inhibitors promote internalization and degradation of ATP1A1 and effectively reduce PEDV infection. (**A**,**B**) The cytotoxicity of Ouabain and PST2238. IPEC-J2 cells were treated with 1.0, 5.0, 10.0, 20.0, 40.0, 80.0 nM Ouabain and 1.0, 5.0, 10.0, 20.0, 40.0, 80.0 μM PST2238 for 48 h; DMSO was used as control. The treated cells were then analyzed using the CCK-8 kit. Data represent means ± SD from three independent experiments. ns, no significant difference. (**C**) Treatment with Ouabain or PST2238 downregulates PEDV RNA abundance. IPEC-J2 cells were grown to 90% confluency, and 1.0 nM Ouabain or 1.0 μM PST2238 viral medium (0.1 MOI) was used to infect the cells for 1 h. After washing the cells with PBS, the maintenance solution containing the above concentrations of drugs was added to continue the culture. Cells were collected at 12, 24 and 48 h for viral RNA quantification. Data represent means ± SD from three independent experiments. *, *p* < 0.05; **, *p* < 0.01; ***, *p* < 0.001. (**D**) Treatment with Ouabain or PST2238 suppresses PEDV RNA abundance in a dose-dependent manner. Virus cultures containing 0.5, 1.0 nM Ouabain or 0.5, 1.0 μM PST2238 were infected with cells for 1 h. The virus-containing medium (0.1 MOI) was replaced with a growth medium containing the corresponding concentration of drugs for 24 h. Extraction of cellular RNA for quantitative PCR assay. *, *p* < 0.05; ***, *p* < 0.001. (**E**,**F**) Treatment with Ouabain or PST2238 decreases PEDV N protein expression in a dose-dependent manner. Cells were treated in the same way as in Figure 4D and then cultured for 48 h. Cells lysates were prepared for Western blot analysis. (**G**) Treatment with Ouabain or PST2238 inhibits PEDV replication in a dose-dependent manner. Cells were treated in the same way as in Figure 4D and then cultured for 48 h. After fixation, penetration and closure of cells, staining with anti-PEDV-N mAb (green) and cell nuclei using DAPI (blue) staining solution. The processed samples were photographed and analyzed using fluorescence microscopy. Scale bars, 100 μm.

**Figure 5 ijms-24-04000-f005:**
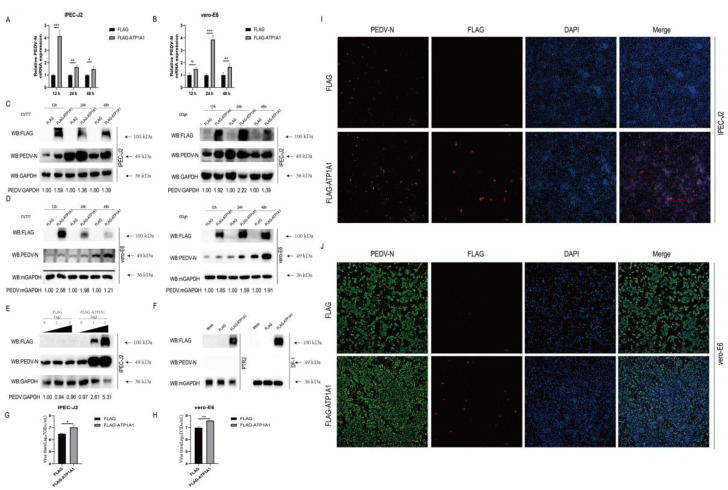
Overexpression of ATP1A1 promotes PEDV infection. (**A**,**B**) Overexpression of ATP1A1 significantly upregulates PEDV RNA abundance. Transfected with FLAG or FLAG-ATP1A1 vector for 24 h, cells were collected at 12, 24 and 48 h after PEDV at 0.1 MOI infection of target cells for quantitative PCR assay. Data represent means ± SD from three independent experiments. *, *p* < 0.05; **, *p* < 0.01; ***, *p* < 0.001; ns, no significant difference. (**C**,**D**) Overexpression of ATP1A1 remarkably facilitates the PEDV N protein expression. After overexpressing of ATP1A1 for 24 h, target cells were infected with PEDV (CV777-G1 or GDgh-G2) at 0.1 MOI and collected and lysed at different time points for Western blot analysis. (**E**) ATP1A1 promotes the expression of PEDV N protein in a dose-dependent manner. Cells were transfected with increasing concentrations of a vector expressing FLAG-ATP1A1. At 24 h post-transfections, the cells were infected with PEDV at 0.1 MOI, and the cell lysates were collected for analysis of PEDV N protein expression with Western blotting. (**F**) PTR2 and DF-1 cells overexpressing of ATP1A1 gene maintained resistance to PEDV infection. Cells overexpressed with ATP1A1 were infected with PEDV at 0.1 MOI for 24 h, and the cell lysates were collected for analysis of PEDV N protein expression with Western blotting. (**G**,**H**) Overexpression of ATP1A1 enhances PEDV viral titers. Cells overexpressed with ATP1A1 were infected with PEDV at 0.1 MOI infected for 24 h, and cell supernatants were extracted for TCID_50_ assay. Data represent means ± SD from three independent experiments. *, *p* < 0.05; **, *p* < 0.01. (**I**,**J**) Overexpression of ATP1A1 enhances PEDV infection. Cells were transfected with FLAG or FLAG-ATP1A1 vector for 24 h, and then PEDV (0.1 MOI) infected for 24 h. The cells were stained with anti-ATP1A1 mAb (red) and anti-PEDV-N mAb (green). Staining of cell nuclei using DAPI (blue) staining solution. The processed samples were photographed and analyzed using fluorescence microscopy. Scale bars, 200 μm.

**Figure 6 ijms-24-04000-f006:**
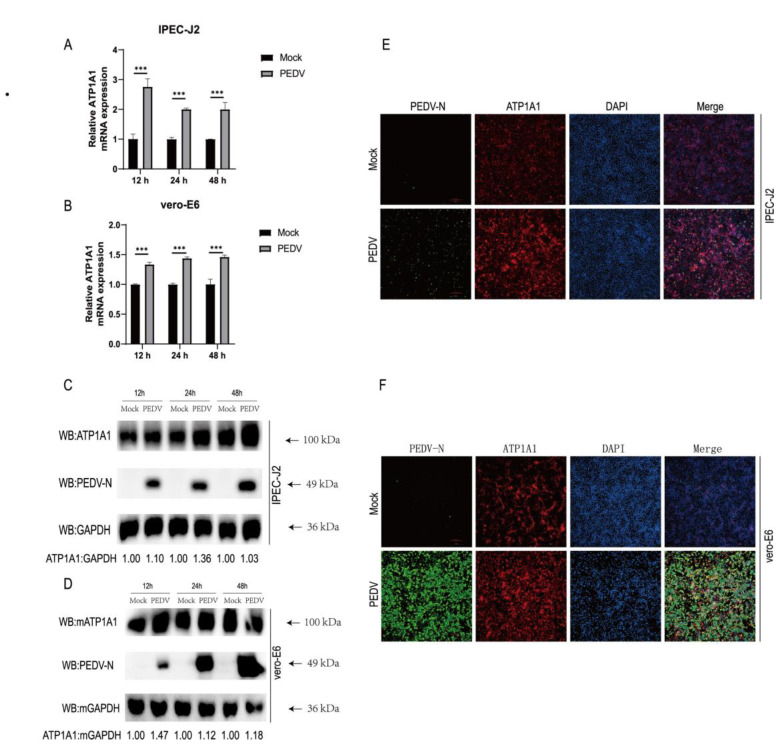
PEDV infection upregulates ATP1A1 protein expression in target cells. (**A,B**) PEDV infection upregulates ATP1A1 mRNA levels. IPEC-J2 and Vero-E6 cells were infected with PEDV at 0.1 MOI. At 12, 24 and 48 h, RNA was extracted, and quantitative PCR analysis performed. Data represent means ± SD from three independent experiments. ***, *p* < 0.001. (**C**,**D**) PEDV infection upregulates ATP1A1 protein expression. PEDV at 0.1 MOI infected with IPEC-J2 and Vero-E6 cells, cells were collected at the indicated time points for Western blot assay. (**E**,**F**) PEDV infection upregulates ATP1A1 protein expression in target cells. The cells were infected with PEDV at 0.1 MOI for 24 h. Cells were stained using triple-color immunofluorescence after fixation and analyzed using fluorescence microscopy. Scale bars, 200 μm.

**Figure 7 ijms-24-04000-f007:**
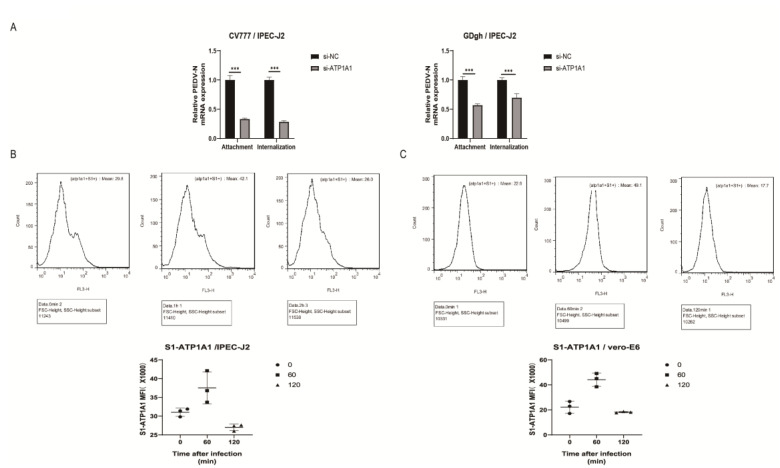
Knockdown of ATP1A1 affects the attachment of PEDV. (**A**) Knockdown of ATP1A1 significantly suppresses attached PEDV. Transfection with siRNAs against ATP1A1 and siRNA-NC for 24 h. In the attachment assay, PEDV at 1.0 MOI infected target cells at 4 °C for 2 h and then cells were washed with PBS and were collected to detect viral RNA abundance. In the internalization assay, PEDV at 1.0 MOI infected target cells at 4 °C for 2 h and then cells were washed with PBS and transferred to 37 °C for 1 h incubation to complete virus internalization. After washing the cells with PBS, the cells were treated sequentially with 0.05% trypsin and 0.5 mg/mL proteinase K to remove the uninternalized virus particles, and the cells were collected for viral RNA detection. Data represent means ± SD from three independent experiments. ***, *p* < 0.001. (**B**,**C**) S1 number of positive cells bound to ATP1A1. Incubating IPEC-J2 and Vero-E6 on ice with PEDV at 1.0 MOI for 1 h to synchronize infection, then transferring to 37 °C incubator for the indicated time. Cells were collected after washing with PBS at the appropriate time points. Cells were then stained using rabbit anti-ATP1A1 mAb and mouse anti-S1 mAb, followed by CoraLite488-conjugated Goat Anti-Rabbit IgG and CoraLite594-conjugated Goat Anti-Mouse IgG. Data analysis using flow cytometry.

**Figure 8 ijms-24-04000-f008:**
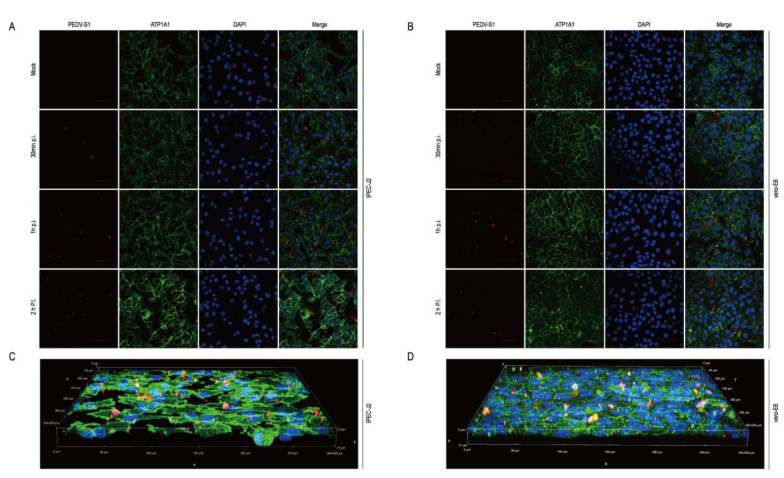
The host protein ATP1A1 co-localizes with the PEDV S1 protein early in PEDV infection. (**A**,**B**) ATP1A1 co-localized with the S1 protein. PEDV at 0.1 MOI infected with IPEC-J2 and Vero-E6 to the specified time. Cells were then stained with antibodies corresponding to Figure 7B. The nuclei were stained with DAPI (blue). The red arrows indicate co-localized signals (scale bar = 50 μm). (**C**,**D**) 3D Rendered Image. PEDV at 0.1 MOI infected IPEC-J2 and Vero-E6 cells for 1 h; cells were stained with the antibodies, with the antibody corresponding to Figure 7B. Staining of cell nuclei using DAPI (blue) staining solution. The panel shows a three-dimensional rendering; the red arrows indicate co-localized signals.

**Figure 9 ijms-24-04000-f009:**
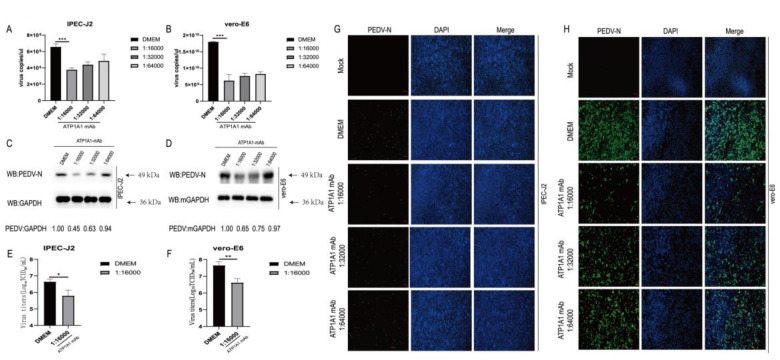
Monoclonal antibody treatment of ATP1A1 effectively reduces PEDV attachment. (**A**,**B**) Treatment with ATP1A1 mAb effectively reduces PEDV RNA abundance. Cells were incubated with serial diluted ATP1A1 mAb at 37 °C for 1 h. Then, the cells were washed with PBS and incubated with PEDV at 0.1 MOI at 4 °C for 1 h. After three washes with PBS, the cells were again incubated with the corresponding mAb in DMEM at 37 °C for 24 h. PEDV RNA abundance was determined by RT-qPCR. Data represent means ± SD from three independent experiments. ***, *p* < 0.001. (**C**,**D**) ATP1A1 mAb treatment significantly decreases PEDV N protein expression. Pretreatment with different folds of diluted ATP1A1 mAb for 1 h, PEDV at 0.1 MOI infected targeted cells for 1 h. Incubation was continued for 48 h in medium containing the corresponding dilution of antibody. Cells were lysed for Western blot analysis. (**E**,**F**) ATP1A1 mAb decreases PEDV viral titers. The experimental processing steps were consistent with Figure 8C. At 48 h, virus yields were determined by TCID_50_ assay with Vero-E6 cells. *, *p* < 0.05; **, *p* < 0.01. (**G**,**H**) ATP1A1 mAb inhibits PEDV attachment in targeted cells. The experimental processing steps were consistent with Figure 8C. PEDV-infected cells were determined by immunofluorescence staining with anti-PEDV-N mAb (green). Staining of cell nuclei using DAPI (blue) staining solution. Scale bars, 100 μm.

**Figure 10 ijms-24-04000-f010:**
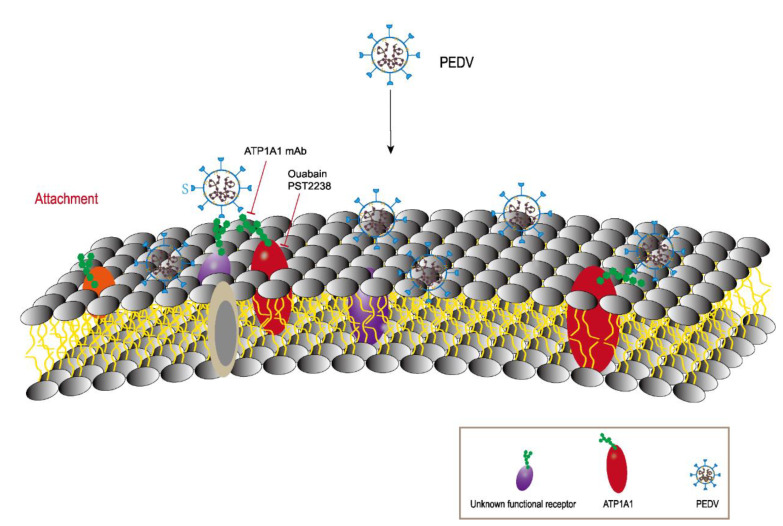
A model describing the involvement of ATP1A1 in PEDV attachment. ATP1A1 plays an important role as a host factor that facilitates PEDV attachment. Overall, ATP1A1 is involved in recognition of hosts together with unknown functional receptors. ATP1A1 mAb interferes with the recognition of ATP1A1 and PEDV on the cell surface and inhibits viral infection. ATP1A1-specific inhibitors treatment leads to partial internalization and degradation of ATP1A1, reducing viral recognition.

**Table 1 ijms-24-04000-t001:** PCR primers used for this study.

Primer	Sequence (5′-3′)
ATP1A1-F ^a^	CGG**GGTACC**ATGGGGAAGGGGGTTGGAC
ATP1A1-CT-F ^a^	CGG**GGTACC**ATGCTGGATGACAACTTCGCCTC
ATP1A1-1R ^a^	CGGGATATCCTAGGCTTGCTGAGGAACCATGTTTTTG
ATP1A1-2R _a_	CGGGATATCCTACACTGCCCTGTTACAAAGACCTG
ATP1A1-3R ^a^	CGGGATATCCTAGGCATCCCTGGGGTTCACC
ATP1A1-4R ^a^	CGGGATATCCTAGACGGTGCCCAGGGG
ATP1A1-R ^a^	CGGGATATCCTAGTAGTAGGTTTCCTTCTCCACCC
PEDV-N-F ^b^	ATGATCTGGTGGCTGCTGTC
PEDV-N-R ^b^	CTTCGAAGTGGCCCTGGATT
Porcine-ATP1A1-F ^b^	CTTGAGCCGAGGCTTAACAC
Porcine-ATP1A1-R ^b^	GAATGCCATAGGCCAAGAAA
Monkey-ATP1A1-F ^b^	CAGCAGTGGACCTATGAGCA
Monkey-ATP1A1-R ^b^	CATTCCAGGGCAGTAGGAAA
Porcine-GAPDH-F ^b^	GATGCTGGTGCTGAGTATGT
Porcine-GAPDH-R ^b^	GGCAGAGATGATGACCCTTT
Monkey-GAPDH-F ^b^	CGAGATCCCTCCAAAATCAA
Monkey-GAPDH-R ^b^	TGACGATCTTGAGGCTGTTG

^a^ The target genes were amplified by PCR and cloned into the KpnI (**bold**) and EcoRV (red) sites in the pECMV-3×FLAG-N vector. ^b^ Used for relative quantitative PCR.

**Table 2 ijms-24-04000-t002:** siRNAs used in this study.

Target Genes	(5′-3′) (Sense)	(5′-3′) (Antisense)
si-NC si-ATP1A1-A si-ATP1A1-B si-mNC si-mATP1A1-A si-mATP1A1-B	UUCUCCGAACGUGUCACGUTT CAGGAAGAACUGCCUUGUGAATT GCAGCUGGAUGACAUCUUGAATT UUCUCCGAACGUGUCACGUTT GAUUCGAAAUGGUGAGAAATT GUGAAGGAGAUGAGAGAAATT	ACGUGACACGUUCGGAGAATT UUCACAAGGCAGUUCUUCCUGTT UUCAAGAUGUCAUCCAGCUGCTT ACGUGACACGUUCGGAGAATT UUUCUCACCAUUUCGAAUCT UUUCUCUCAUCUCCUUCACTT

## Data Availability

Data Availability Statements are available in section “MDPI Research Data Policies” at https://www.mdpi.com/ethics.

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
