# Peer review of "The Alpha-1 Subunit of the Na+/K+-ATPase (ATP1A1) Is a Host Factor Involved in the Attachment of Porcine Epidemic Diarrhea Virus"

_ijms, 2023, doi:10.3390/ijms24044000_

Round 1

Reviewer 1 Report

1, lines 43-53. The study progress is not fully described, integrin αvβ1 is also its auxiliary entry receptor, however it is not mentioned.
2, The article has many formatting problems and needs further checking, for example: line 433 CO2 , line 527 4°C, References, etc.
3、Fig7B picture is not clear.

Reviewer 2 Report

In this report, Xiong et al are examining the role of ATP1A1 on PEDV virus infection. Previous reports have identified the pAPN to be the main receptor of PEDV entry. However, a number of contrasting reports challenging the role of pAPN on PEDV entry have emerged. The authors, after performing a VOPBA assay, they found that ATP1A1 is an important entry factor that interacts with PEDV S1 utilizing an unknown mechanism. The authors have performed a number of assays showing that disruption of ATP1A1 results in the reduction of virus infection. Nevertheless, the manuscript has some important weaknesses that need to be addressed. Some of the experiments lack some important controls or they are interpreted incorrectly. The manuscript would benefit immensely from a native English speaker, as some sentences in the abstract and introduction are confusing and unclear. The authors use repeatedly knockdown assays with siRNAs specific to ATP1A1, the knockdown is good but makes the data not very clean. It would be better to make a KO cell line by CRISPR/Cas9.  Finally, the way some experiments are laid out do not necessarily address the questions the authors are making.

Issues with the use of English in the abstract and introduction need to be addressed. E.g. “ ATP1A1 was top one binding to …” in line 14, is unclear; top one… what do the authors mean by that. Issues and phrases like that persist especially in the first two sections of the manuscript.

Lines 51-52, to claim that pAPN is questionable as a receptor is a little premature. It is possible that receptors for PEDV may be cell line specific. The authors should rephrase that.

In line 57, “..antibody (receptor protein) in vitro” is not clear. What do they mean by receptor protein? Please rewrite.

In all immunoblots, molecular weights are missing and should be included.

In figure 1A, the authors need to include a control where they have knocked down by siRNA the ATP1A1 gene and then infect the cells. This negative control is essential to show that this is a specific interaction.  Also, where is the reciprocal coIP? Pulling down with ATP1A1 and probing for S1.

In figure 1C, what do the arrows point at? This is never explained

In figure 1C, where is S1 in the input? Also why are the levels of S1 in the coIP, when pulling down with S1, are decreased in lanes 2 and 3? This needs to be addressed or repeated.

In line 119, “the double strand of siRNA-ATP1A1” is confusing. What does that mean? Please rewrite.

Figure 2F the simATP1A1-B protein levels are not agreeing with the RNA levels shown in 2D. It shows that there is no effect. This is very strange, as the RNA levels are abrogated. The authors need to repeat this experiment.

The authors are using siRNA knockdown but the effect from the knockdown on virus infection is not very pronounced. So this physiologically important? It would be “cleaner” if the authors repeated the experiments in figure 3 at least using actual complete knockout cell lines by using CRISPR/Cas9 and using pseudoviruses.

In figure 3 and some of the other figures, the authors do not provide all experimental details. For example how much viruses is being used for infections in Figure 3? Please provide that information. What where the antibody dilutions in your western blots? Please provide that information.

In line 136, the authors state that they see the effect “especially at 12h”. This reviewer does not understand why just at 12 h, as a similar effect is seen at the other 2 time points. The authors need to correct this.

The authors state that the effect of ATP1A1 on PEDV is seen in both the IPECJ2 and Veros. However, the effect in Veros at the PDEV protein levels as seen in 3D is not really observed. The protein levels of the virus in the Veros whether you have knockdown or not, are similar. The difference at best is negligible. If they really want to say something about protein levels, quantification of their immunoblots should be performed. The data in 3D are not really convincing. The same is true for the GDgh-G2 strain for the most part, the data are not very convincing, like before, the authors should quantify the bands in their blots.

The authors, when referring to figures 3G and H, they say there is a “significant reduction” (line 140) in PDEV fluorescence. The authors need to quantify the signal in their images and provide a quantitative analysis if they want to say anything about “significant reduction” in signal. A more (better) quantitative analysis would be to perform flow cytometry of stained cells to get a more of a quantitative understanding.      

It is not clear why in Figure 4C, they only observed an effect at 24hours, as in the previous figure (Fig 3) the effect was seen as early as 12hours and lasted to the end of the experiment. This is perplexing. Why is that? The authors should try to change their conditions using the drugs, because the data are just not convincing. Also, the authors are looking as late as 72 hours for drug toxicity, but their infections do not go past 48hours. In that case, they should just look at toxicity levels up to 48 hours, as what happens later on is irrelevant to their infection model. Also why did they not see an effect at 12h and 48h? This graph is confusing.

In figure 4E and F, they examine what happens at 48 hours, yet in Figure 4C, they showed no effect at 48 hours. Why is there such discrepancy?

The images in Figure 4G are not very clear or convincing. A quantitative approach would be more appropriate.

In line 214, the authors state “significantly lower”, but in the data they show that the levels are higher. Please correct this.

The statement “in addition, we attempted to overexpress….(data not shown)” (line 220-221) is very important and the data should be shown.

The data shown in Fig. 6C and D are just not convincing. There is not apparent difference in the protein levels. The blots look like they are past linearity. The authors should provide titer data to show that there is more virus produced possibly due to enhanced virus infectivity of the cells overexpressing ATP1A1.

The authors state that their immunofluorescence experiments show that ATP1A1 is in the cell membrane (line 254). However, their data never show that. In figure 6E and F they do not provide any information that would even suggest that. To talk about membrane localization, the authors need to do fractionations of cell membranes from cells and then perform immunoblots to detect their presence there, or at least some FACS experiments with surface staining to show that ATP1A1 is found on the cell surface. That statement is incorrect and is not addressed in the experiments they have performed. This experiment and interpretation is very problematic.

The authors state that the experiment in Figure 7 shows that ATP1A1 shows increased attachment of virus to cells. However, the experiment they have done does not really show that. The readthrough is viral RNA in Fig 7A, which is not correct, as the RNA they are seeing they do not know if this is from input virus or from virus replicating in the cell, as then they can’t rule out the effect of ATP1A1 on virus entry or post entry events. Also incubating the virus for 1h at 37οC allows for the virus to also be internalized by endocytosis so again this does not necessarily show attachment. They should label virions with FITC or another fluorochrome and then perform flow cytometry for FITC to show that the label is still there (or it has changed) on the surface of the cells. That will say if attachment is affected.

If the authors think that internalization is unaffected and the effect they see is due to attachment, then they should normalize the results of the internalization experiments with their attachment results. Otherwise, they can’t rule out that ATP1A1 is affecting also internalization.  

In figure 7b and c does not show an association between ATP1A1 and PEDV S1. They just that these cell also express ATP1A1. They need to include an additional control where they knockdown ATP1A1 to show that these two proteins are interacting.

In figure 8, to show colocalization, the authors need to perform Z-stack analysis. The way they have analyzed their data does not prove that S1 and ATP1A1 colocalize. Furthermore, they need to quantify colocalization by determining the colocalization coefficiency. Also a negative control (knockdown of ATP1A1) is missing.

Figure 9 shows attachment rather than infection as it is kept at 4 οC. They should change the title of the figure from PEDV infection to PEDV attachment.

Lines 372-373 make no sense, they need to be rewritten.

In their discussion, the authors need to emphasize that ATP1A1 enhances entry and should avoid any possible mention that this could be the receptor. The fact that cell lines (DF1) that overexpress it and still are resistant to infection suggest most likely that this is not a receptor but possibly an entry cofactor. Furthermore, a lot of the data here show a very modest effect, thus it further emphasizes that this protein is possibly just a host factor that assists virus entry.

 It is also not clear why the authors never did pseudovirus experiments. Pseudoviruses are ideal for the questions they are asking as they do not undergo multiple rounds of infection, which is a problem with infectious viruses. Therefore, you get a better understanding on the effect of the host gene on virus entry. This reviewer thinks that they should at least perform a few experiments showing the effect of this gene on PEDV entry via pseudoviruses.

Round 2

Reviewer 1 Report

1. The authors listed the role of integrin αvβ1 in the process of various viral infections, but did not find a role in PEDV or coronavirus in the process of viral infection, indicating unfamiliarity with the progress of research on PEDV. The following literature can be consulted12.

2. Fig 6.C PEDV-N picture is not a whole, please re-upload the picture. Also the images in Fig1 are too bad, especially Fig1D input WB-S1. It is suggested to re-map most WB results.

1Davis Paul J,Lin Hung-Yun,Hercbergs Aleck et al. Coronaviruses and Integrin αvβ3: Does Thyroid Hormone Modify the Relationship?[J] .Endocr Res, 2020, 45: 210-215.

2Li Chunqiu,Su Mingjun,Yin Baishuang et al. Integrin αvβ3 enhances replication of porcine epidemic diarrhea virus on Vero E6 and porcine intestinal epithelial cells.[J] .Vet Microbiol, 2019, 237: 108400.

Reviewer 2 Report

This is a revised version of the manuscript regarding the role of ATP1A1 on PEDV infection. The authors, more accurately, now described it as an entry/attachment factor, which makes sense. Many previous issues have been resolved. However, some remain. The biggest issue that really affects the whole paper is clarity. The paper would truly benefit from an English speaker going through it.

English language issues persist in the abstract and introduction.

In line 33, it should be “positive sense” not positive stranded.

Lines 33-37 is a large run on sentence that makes it hard to understand. Please rewrite.

Line 38 is not clear. What encodes 4 different structural proteins. It needs to be rewritten.

Line 39, what does mediated by a trimer mean? It is unclear.

Line 117-119, “different siRNAs…expression”. This sentence needs to be rewritten for clarity.

Figure 2F, the GAPDH blot is identical to the GAPDH blot from the 2F figure from the previous version of the paper, yet the ATP1A1 is from a new blot that is markedly different from the ATP1A1 blot shown in the previous version of the paper. This is unacceptable. The loading control blot should be from the blot of ATP1A1 presented here.

The statement in line 137-138 that there was a dramatic decrease in N levels in VeroE6, even after quantification, it is a very modest decrease at best, whose biological significance is questionable. The authors need to fix that, as this is an exaggeration of the findings. The same is true for the GDghG2 strain for the most part, the data are not very convincing. It is possible that ATP1A1 is acting in a strain and cell type specific manner. The authors need to mention something of that effect in their manuscript.

In figure 4C, at 12h, are the authors comparing Ouabain to PST or Ouabain to DMSO? The bar with the two stars over the columns seems to be misplaced.

In figure 6C, the PEDV-N blot here is very different from the previous version of the paper. Yet, the blots for ATP1A1 and GAPDH are the same. Is that the same experiment? They do not look like these bands are from the same experiment. The blots for ATP1A1 and GAPDH should be from the same experiment with PEDV-N blot, otherwise this is troubling. It also looks that the blot for the lanes indicating 12, 24 and 48 hours were individually adjusted, which is also troubling.

The issue in 6D persists, there seem to be no marked increase of ATP1A1 in the presence of infection at any of the time points. When comparing mock to PEDV, only at 12 hours there is a difference. At 24 and 48, no difference is observed.

The authors cant claim in line 255-256 that “is commonly used as a marker for membrane proteins(32).”, when they cite only 1 paper. There needs to be multiple papers for them to use such a phrase in their manuscript.

The authors state in lines 263-265 that “ATP1A1 protein shifted from uniform distribution in the cell membrane to filling the cytoplasm in large amounts compared with mock-treated Vero-E6 cell”. However, in this low magnification image they cant really say that. Yes, the signal is more enhanced but to talk about changes in localization, they need to do Z-stacking analysis. What they see in this image could be high amounts of the protein on the surface of the cells and not redistribution to cytosol. Again fractionation experiments would be the way to do it.

If the authors, cant perform Z-stacking analysis to talk about S1 and ATP1A1 colocalization, then they should not be presenting that data as the data do not show colocalization. The 2.8 section and figure 8 should be removed from the manuscript as the conclusions drawn from it are not scientifically sound.

The authors need to change the title of figure 9 so the new title to says “attachment” and not infection.

Round 3

Reviewer 2 Report

The paper is significantly improved in most places. Some issues with English persist (two examples: in line 270 "to further understanding" should be "to further understand" and line 66-67 is a sentence with no verb!). Please hire someone to correct the English, because there are parts of the paper that are difficult to understand. 

Author Response

We have done our best to revise the manuscript and ask reviewers to check it in the revised version.